# A Bi-level Framework for Debiasing Implicit Feedback with Low Variance

## Abstract

Implicit feedback is easy to collect and contains rich weak supervision signals, thus is broadly used in recommender systems. Recent works reveal a huge gap between the implicit feedback and the user-item relevance due to the fact that users tend to access items with high exposures but these items may not be necessarily relevant to users' preferences. To bridge the gap, existing methods explicitly model the item exposure degree and propose unbiased estimators to improve the relevance. Unfortunately, these unbiased estimators suffer from the high gradient variance, especially for long-tail items, leading to inaccurate gradient updates and degraded model performance.

To tackle this challenge, we propose a bi-level framework for debiasing implicit feedback with low variance. We first develop a low-variance unbiased estimator from a probabilistic perspective, which effectively bounds the variance of the gradient. Unlike previous works which either estimate the exposure via heuristic-based strategies or use a large biased training set, we propose to estimate the exposure via an unbiased small-scale validation set. Specifically, we parameterize the user-item exposure by incorporating both user and item information, and propose to construct the unbiased validation set only from the biased training set instead of using random policy at the cost of degrading user experience. By leveraging the unbiased validation set, we adopt a bi-level optimization framework to automatically update exposure-related parameters along with recommendation model parameters during the learning. Experiments on two real-world datasets and two semi-synthetic datasets verify the effectiveness of our method. Our code is available at `https://anonymous.4open.science/r/TMLR-Biff/README.md`.

## 1 INTRODUCTION

Recent years have witnessed the fast development of the recommender system. It has been successfully deployed in many web services like E-commerce and social media. Learning from historical interactions, a recommender system can predict the relevance or preference between users and items, based on which, the system recommends items that the user may prefer. To enable these, there are two types of feedback: explicit feedback and implicit feedback. Explicit feedback can be the ratings on items that explicitly represent the preferences of the users. However, collecting explicit feedback requires the user active participation, which makes explicit feedback unavailable in most real-world scenarios. Compared with explicit feedback, implicit feedback such as clicks is widely used because of its ubiquity and wide availability. Though easier to collect, implicit feedback is one-sided and positive only Yang et al. (2018), which means the recommender can only observe the user interactions with relevant items. A missing link between a user and an item can either be that the user dislikes the item or that the item is not exposed to the user Liang et al. (2016b).

Many important works have tried to improve recommendation performances in implicit feedback by explicitly modeling the user-item exposure. For example, the work Yang et al. (2018) finds implicit feedback subject to popularity bias, and proposes an unbiased evaluator based on the Inverse-Propensity-Scoring (IPS) technique Joachims & Swaminathan (2016), which significantly reduces the evaluation bias. Exposure matrix factorization (ExpoMF) Liang et al. (2016b) introduces exposure variables to build a probabilistic

model, and considers external information when estimating exposure. Yet, RelMF Saito et al. (2020) finds that ExpoMF is biased towards popular items and yields unsatisfied results for rare items. Based on the IPS technique, RelMF proposes an unbiased estimator to maximize the user-item relevance. In their work, both the user-item exposure and the user-item relevance are modeled as Bernoulli random variables, and the click probability is the product of the exposure probability and the relevance probability. RelMF better achieves the objective of the unbiased recommendations than alternatives Liang et al. (2016b); Hu et al. (2008). Using the same unbiased estimator in RelMF, CJMF Zhu et al. (2020) adopts a combinational joint learning framework to more accurately estimate exposure.

However, we find Saito et al. (2020); Zhu et al. (2020) suffer from the high gradient variance problem. Inaccurate gradient updates occur in the learning process, which degrades the model performance. Moreover, these existing approaches Hu et al. (2008); Yang et al. (2018); Saito et al. (2020); Zhu et al. (2020) adopt some simple heuristic-based strategies or only leverage the biased training set to estimate exposure, which inevitably leads to a biased recommendation model.

To tackle the high gradient variance problem, we develop a low-variance estimator from a probabilistic perspective. To better estimate exposure, we model exposure by incorporating both user and item information and propose to construct a small-scale unbiased validation set only from the biased training set to guide exposure estimation. This construction process does not involve any random policy and thus will not degrade user experience. With the unbiased set, we introduce bi-level optimization Colson et al. (2007) with exposure parameters as the outer variable and relevance parameters (recommendation model parameters) as the inner variable, to update exposure parameters automatically. Overall, we propose **Biif**: *A **Bi**level framework for Debiasing **I**mplicit **F**eedback with Low Variance* to update exposure parameters simultaneously with relevance parameters. We further analyze the inner mechanism of the bi-level framework in Biif and compare bi-level optimization with other optimization methods to demonstrate the necessity of the bi-level framework. We verify the effectiveness of Biif on both real-world and semi-synthetic datasets.

To summarize, our work has three contributions:

1. We propose a low-variance unbiased estimator which effectively bounds gradient variance.

2. We connect exposure estimation to both user and item information and propose to form a unbiased set without degrading user experience. With the unbiased set, we propose a bi-level framework to guide exposure estimation.

3. Furthermore, we give a natural interpretation of why bi-level optimization works by gradient analysis, and compare it with other optimization methods to better understand its necessity.

The structure of the paper is as follows. In Section 2, we introduce some notations and illustrate the Biif algorithm. Then we analyze the experimental results on two real-world datasets in Section 3 and two semi-synthetic datasets in Section 4. In Section 5, we discuss related work from two perspectives: debiasing and bi-level optimization in the recommender system. In Section 6, we summarize our findings.

## 2 METHOD

In this section, we begin by introducing some preliminaries including notations and the previous unbiased estimator. Then we show the high gradient variance problem in the previous unbiased estimator and derive our low-variance unbiased estimator from a probabilistic perspective. Further, we parameterize the user-item exposure by considering both user and item information and propose to construct a small unbiased validation set to guide exposure estimation via a bi-level optimization framework.

### 2.1 Preliminaries

**Notations.** Assume we have an implicit feedback dataset $\mathcal{D}$ with $N$ users indexed by $u$ and $M$ items indexed by $i$. Let $\tilde{R}_{ui}$ denote the observed feedback between $u$ and $i$. $\tilde{R}_{ui} = 1$ indicates positive feedback, while $\tilde{R}_{ui} = 0$ indicates either positive unlabeled feedback or negative feedback.

To precisely formulate implicit feedback, RelMF Saito et al. (2020) introduces two kinds of Bernoulli random variables $R_{ui}$ and $O_{ui}$. $R_{ui}$ represents the user-item relevance between $u$ and $i$ with $\gamma_{ui}$ as the Bernoulli parameter. $R_{ui} = 1$ means $u$ and $i$ are relevant, and $R_{ui} = 0$ means $u$ and $i$ are not relevant. Similarily, $O_{ui}$ represents the user-item exposure between $u$ and $i$ with $m_{ui}$ as the Bernoulli parameter. $O_{ui} = 1$ means $i$ is exposed to $u$, and vice versa. We denote $\bar{m}_{ui}$ as the estimated exposure between $u$ and $i$ in the following paper. Note that both $R_{ui}$ and $O_{ui}$ can not be observed in implicit feedback. $\tilde{R}_{ui}$ is also a Bernoulli variable:

$$\tilde{R}_{ui} = R_{ui} O_{ui}. \tag{1}$$

The Bernoulli parameter of $\tilde{R}_{ui}$ can be written as:

$$P(\tilde{R}_{ui} = 1) = m_{ui} \gamma_{ui}. \tag{2}$$

From Eq. (1), we can see that a positive feedback $R_{ui} = 1$ means that $i$ is exposed to $u$ and $u$ likes $i$.

The task of the implicit recommendation system is to provide an ordered set of items for users based on the predicted user-item relevance. We use $p_{ui} = p(R_{ui} = 1|\boldsymbol{\omega})$ to represent the predicted user-item relevance where the relevance parameters (recommendation model parameters) $\boldsymbol{\omega}$ include the user embedding $\boldsymbol{\omega_u}$ and the item embedding $\boldsymbol{\omega_i}$. Since matrix factorization is the most widely used technique Koren et al. (2009) in the recommender system, in this paper we compute the predicted user-item relevance as:

$$p_{ui} = \sigma(\boldsymbol{\omega_u}^\top \boldsymbol{\omega_i}), \tag{3}$$

where $\sigma(\cdot)$ represents the sigmoid function. Note that our Biif algorithm can also be easily applied on other neural network based modelsHe et al. (2017); Wang et al. (2019a).

**Unbiased Estimator.** The work Saito et al. (2020) finds the top-k recommendation metrics such as the mean average precision Yang et al. (2018) can not directly signify relevance, and thus are not proper to measure recommendation results. To optimize the performance metric of relevance, the work Saito et al. (2020) proposes an unbiased estimator from the IPS technique, and the log loss form can be written as:

$$\mathcal{L}_1(\boldsymbol{\omega}) = -\sum_{(u,i)\in\mathcal{D}} \frac{\tilde{R}_{ui}}{\bar{m}_{ui}} \log p_{ui}$$
$$+ (1 - \frac{\tilde{R}_{ui}}{\bar{m}_{ui}}) \log(1 - p_{ui}). \tag{4}$$

Once we have the expectation of $\mathcal{L}_1(\boldsymbol{\omega})$, we will find the optimal solution for $p_{ui}$ is $\gamma_{ui}$ given an accurate exposure estimation $\bar{m}_{ui} = m_{ui}$. This proves this estimator unbiased. In this paper, we mainly consider the log loss form since it is the most widely used form. Other loss forms such as the mean squared loss can be analyzed similarly.

## 2.2 Proposed Unbiased Estimator

**High gradient variance**. We compute the gradient of $\mathcal{L}_1(\boldsymbol{\omega})$ with respect to $p_{ui}$ as:

$$\frac{\partial \mathcal{L}_1(\boldsymbol{\omega})}{\partial p_{ui}} = -[\frac{\tilde{R}_{ui}}{\bar{m}_{ui}}(\frac{1}{p_{ui}} + \frac{1}{1-p_{ui}}) - \frac{1}{1-p_{ui}}]. \tag{5}$$

The variance of $\frac{\partial \mathcal{L}_1(\boldsymbol{\omega})}{\partial p_{ui}}$ can be calculated by:

$$V(\frac{\partial \mathcal{L}_1(\boldsymbol{\omega})}{\partial p_{ui}}) = \frac{V(\tilde{R}_{ui})}{\bar{m}_{ui}^2}(\frac{1}{p_{ui}} + \frac{1}{1-p_{ui}})^2$$
$$= \frac{\gamma_{ui}(1 - \bar{m}_{ui}\gamma_{ui})}{\bar{m}_{ui}p_{ui}^2(1-p_{ui})^2}. \tag{6}$$

For rare items, $\bar{m}_{ui}$ can be very small so that $V(\frac{\partial \mathcal{L}_1(\boldsymbol{\omega})}{\partial p_{ui}})$ becomes unbounded. This problem leads to inaccurate gradient updates and decreases the model performance.

**Low-variance unbiased estimator**. Instead of deriving from the IPS technique, which leads to the high gradient variance problem, we propose a low-variance unbiased estimator from a probalistic view. More specifically, we first write the cross-entropy loss as:

$$\mathcal{L}(\boldsymbol{\omega}) = - \sum_{(u,i)\in\mathcal{D}} \tilde{R}_{ui} \log p(\tilde{R}_{ui} = 1|\boldsymbol{\omega})$$
$$+ (1 - \tilde{R}_{ui}) \log p(\tilde{R}_{ui} = 0|\boldsymbol{\omega}). \tag{7}$$

Recall that we are caring about the user-item relevance prediction $p_{ui}$. From the probabilistic perspective, we could have:

$$p(\tilde{R}_{ui} = 1|\boldsymbol{\omega}) = \bar{m}_{ui} p_{ui}. \tag{8}$$

$$p(\tilde{R}_{ui} = 0|\boldsymbol{\omega}) = 1 - \bar{m}_{ui} p_{ui}. \tag{9}$$

Our estimator is defined as:

$$\mathcal{L}_2(\boldsymbol{\omega}) = - \sum_{(u,i)\in\mathcal{D}} \tilde{R}_{ui} \log(\bar{m}_{ui} p_{ui})$$
$$+ (1 - \tilde{R}_{ui}) \log(1 - \bar{m}_{ui} p_{ui}). \tag{10}$$

After computing the expectation of $\mathcal{L}_2(\boldsymbol{\omega})$, we can easily find the optimal solution for $p_{ui}$ is also $\gamma_{ui}$ given an accurate exposure estimation $\bar{m}_{ui} = m_{ui}$. This proves our estimator unbiased (see Appendix A.1 for details). Besides, our unbiased estimator yields better gradient properties for rare items. Specifically, we calculate the gradient as:

$$\frac{\partial \mathcal{L}_2(\boldsymbol{\omega})}{\partial p_{ui}} = -\left(\frac{\tilde{R}_{ui}}{p_{ui}} + \frac{(\tilde{R}_{ui} - 1)\bar{m}_{ui}}{1 - \bar{m}_{ui} p_{ui}}\right). \tag{11}$$

The variance of $\frac{\partial \mathcal{L}_2(\boldsymbol{\omega})}{\partial p_{ui}}$ is calculated as:

$$V\left(\frac{\partial \mathcal{L}_2(\boldsymbol{\omega})}{\partial p_{ui}}\right) = V(\tilde{R}_{ui})\left(\frac{1}{p_{ui}} + \frac{\bar{m}_{ui}}{1 - \bar{m}_{ui} p_{ui}}\right)^2$$
$$= \frac{\bar{m}_{ui} \gamma_{ui}(1 - \bar{m}_{ui} \gamma_{ui})}{p_{ui}^2 (1 - \bar{m}_{ui} p_{ui})^2}. \tag{12}$$

$V\left(\frac{\partial \mathcal{L}_2(\boldsymbol{\omega})}{\partial p_{ui}}\right)$ stays bounded as $\bar{m}_{ui}$ becomes small, and thus this estimator yileds stable gradient updates. We do not consider the possible high gradient variance problem caused by $p_{ui} = 0$ or $p_{ui} = 1$ since this occurs in both estimators. Our estimator only solves the high gradient variance problem related to $\bar{m}_{ui}$.

## 2.3 Exposure Estimation

### 2.3.1 Exposure Modeling

It is not realistic to assign every $O_{ui}$ entry a learnable parameter to represent the user-item exposure due to the space limit, so we need a distributed representation for all $O_{ui}$ entries. In this paper, we parameterize the user-item exposure $m_{ui}$ with one MLP (multi-layer perceptron) and $N$ user-wise embeddings, and connect exposure estimation with both user and item information. On one hand, $m_{ui}$ is large if the item is popular, which means we should consider the item popularity when estimating $m_{ui}$. Note that the popularity He et al. (2016) of the item $i$ can be approximated as the following:

$$\theta_i = \left(\frac{\sum_u \tilde{R}_{ui}}{max_i \sum_u \tilde{R}_{ui}}\right)^{0.5}. \tag{13}$$

On the other hand, $m_{ui}$ becomes large if the item is exposed to the user often or the user is active, which means we should also consider the impact of the user. We introduce a new user-wise embedding $\boldsymbol{e_u}$ and use

---

**Algorithm 1** Bi-level Optimization Framework

---

**Input**: the training set $\mathcal{D}_{train}$; the unbiased validation set $\mathcal{D}_{val}$; the max iteration $T$; other hyperparameters.
**Parameter**: $\boldsymbol{\omega}$: the relevance parameters(the user and the item embeddings); $\boldsymbol{\alpha}$: the exposure parameters to parameterize $\bar{m}_{ui}$.
**Output**: $\boldsymbol{\omega^*}$.

1: **for** $t \in range(0, T)$ **do**
2:    SampleMiniBatch $\mathcal{B}_{train}$ from $\mathcal{D}_{train}$;
3:    Update $\boldsymbol{\omega}$ with $\mathcal{B}_{train}$ by Eq.(16);
4:    Update $\boldsymbol{\alpha}$ with $\mathcal{D}_{val}$ by Eq.(17);
5: **end for**

---

$\sigma(\boldsymbol{e_u}^\top \boldsymbol{\omega_i})$ to represent the user impact. Note that the introduced user embedding $\boldsymbol{e_u}$ can be learned directly through external user information Liang et al. (2016b), whereas, in this paper, we assume we do not have the external information, which is more general. To sum up:

$$m_\alpha(u, i, \boldsymbol{\omega}) = r(\boldsymbol{\omega_i})\sigma(\boldsymbol{e_u}^\top \boldsymbol{\omega_i}) + (1 - r(\boldsymbol{\omega_i}))\theta_i. \tag{14}$$

Here $r(\cdot)$ learns the trade-off between the impact of the user and the popularity of the item, and we use one layer MLP followed by a sigmoid function to parameterize $r(\cdot)$. We use $\alpha$ to denote exposure parameters, which include the introduced user embeddings and the MLP parameters in $r$. For convenience, we still use $\bar{m}_{ui}$ instead of $m_\alpha(u, i, \boldsymbol{\omega})$ to represent the estimated exposure in the following paper.

### 2.3.2 Bi-level Optimization

Previous work Hu et al. (2008); Yang et al. (2018); Saito et al. (2020); Zhu et al. (2020) adopt some simple heuristics or only use the biased training set to estimate exposure, which inevitably results in a biased model. We propose to construct a small unbiased validation set to guide exposure estimation via bi-level optimization. Specifically, we select the most popular positive item and negative item for each active user to form the unbiased validation set. The reason why the validation set can be treated as unbiased is that these items are very likely to be exposed to these users and we approximate $m_{ui}$ as 1 in the validation set. The proposed unbiased validation set is different from the one in AD Chen et al. (2021), which is collected by deploying random policy to the recommendation platform and degrades the user experience a lot. This means the proposed Biff algorithm can be much more practical than AD.

**Formulation**. We use the proposed estimator in Eq. (10) to calculate the training loss $\mathcal{L}_{train}$ and the validation loss $\mathcal{L}_{val}$. Given an unbiased training set, we obtain the optimal user and item embeddings $\boldsymbol{\omega^*}$ by minimizing $\mathcal{L}_{train}(\boldsymbol{\omega})$. Whereas in a biased training set, different user-item pairs have different exposure. To be specific, for a biased training set, we need to first estimate the user-item exposure $\bar{m}_{ui}$ parametrized by $\boldsymbol{\alpha}$. Given $\bar{m}_{ui}$, the optimal $\boldsymbol{\omega}$ is computed as:

$$\boldsymbol{\omega}^*(\boldsymbol{\alpha}) = \arg\min_{\boldsymbol{\omega}} \mathcal{L}_{train}(\boldsymbol{\omega}, \boldsymbol{\alpha}). \tag{15}$$

The exposure parameters $\boldsymbol{\alpha}$ can be seen as a special type of hyper-parameter and we update $\boldsymbol{\alpha}$ automatically by minimizing the validation loss $\mathcal{L}_{val}(\boldsymbol{\omega}^*(\boldsymbol{\alpha}))$ on the unbiased validation set. Note that $\mathcal{L}_{val}(\boldsymbol{\omega}^*(\boldsymbol{\alpha}))$ does not explicitly contain any $\boldsymbol{\alpha}$ term since the user-item exposure $m_{ui}$ is approximated as 1 in the unbiased validation set.

Our formulation implies a bi-level optimization problem with exposure parameters $\boldsymbol{\alpha}$ as the outer variable and the model parameters $\boldsymbol{\omega}$ as the inner variable:

$$\min_{\boldsymbol{\alpha}} \quad \mathcal{L}_{val}(\boldsymbol{\omega}^*(\boldsymbol{\alpha})). \tag{1Biif}$$

$$\text{s.t.} \quad \boldsymbol{\omega}^*(\boldsymbol{\alpha}) = \arg\min_{\boldsymbol{\omega}} \mathcal{L}_{train}(\boldsymbol{\omega}, \boldsymbol{\alpha}). \tag{2Biif}$$

For efficiency, we use a gradient step with the learning rate $\eta$ to approximate $\boldsymbol{\omega}^*(\boldsymbol{\alpha})$ in the inner loop:

$$\boldsymbol{\omega}^*(\boldsymbol{\alpha}) \approx \boldsymbol{\omega} - \eta \frac{\partial \mathcal{L}_{train}(\boldsymbol{\omega}, \boldsymbol{\alpha})}{\partial \boldsymbol{\omega}}. \tag{16}$$

Similarly, in the outer loop, we update $\boldsymbol{\alpha}$ by minimizing $\mathcal{L}_{val}(\boldsymbol{\omega}^*(\boldsymbol{\alpha}))$ via a gradient descent step with the outer loop learning rate $\eta^{'}$:

$$\boldsymbol{\alpha}^* \approx \boldsymbol{\alpha} - \eta^{'} \frac{\partial \mathcal{L}_{val}(\boldsymbol{\omega}^*(\boldsymbol{\alpha}))}{\partial \boldsymbol{\alpha}}. \tag{17}$$

This proposed bi-level optimization framework is summarized in Algorithm 1.

**Interpretation by gradient analysis**. By analyzing gradients, we give a natural interpretation of bi-level optimization in Biif. In the validation set, assume $u$ likes $i_1$ and dislikes $i_2$ ($i_1$ or $i_2$ can not be $i$); $u_1$ likes $i$ and $u_2$ dislikes $i$ ($u_1$ or $u_2$ can not be $u$). We first compute the gradient for $\tilde{R}_{ui} = 1$ (see Appendix A.2):

$$\frac{\partial \mathcal{L}^{'}_{val}}{\partial \bar{m}_{ui}} = 0. \tag{18}$$

This means $\bar{m}_{ui}$ will not be updated explicitly for the positive feedback. Denote $\boldsymbol{\omega_u}^\top \boldsymbol{\omega_i}$ as $\bar{R}_{ui}$ and then we compute the gradient for $\tilde{R}_{ui} = 0$ (see Appendix A.3 for details):

$$\begin{aligned}
\frac{\partial \mathcal{L}^{'}_{val}}{\partial m_{ui}} &= \frac{\partial \mathcal{L}^{'}_{val}}{\partial \boldsymbol{\omega_u}^\top} \frac{\partial \boldsymbol{\omega_u}(m_{ui})}{\partial m_{ui}} + \frac{\partial \mathcal{L}^{'}_{val}}{\partial \boldsymbol{\omega_i}^\top} \frac{\partial \boldsymbol{\omega_i}(m_{ui})}{\partial m_{ui}} \\
&= \frac{\eta \sigma(\bar{R}_{ui}) \sigma(-\bar{R}_{ui})}{(1 - \bar{m}_{ui} \sigma(\bar{R}_{ui}))^2} [\boldsymbol{\omega_{i_1}}^\top \boldsymbol{\omega_i} \sigma(-\bar{R}_{ui_1}) \\
&\quad - \boldsymbol{\omega_{i_2}}^\top \boldsymbol{\omega_i} \sigma(\bar{R}_{ui_2}) + \boldsymbol{\omega_{u_1}}^\top \boldsymbol{\omega_u} \sigma(-\bar{R}_{u_1 i}) \\
&\quad - \boldsymbol{\omega_{u_2}}^\top \boldsymbol{\omega_u} \sigma(\bar{R}_{u_2 i})].
\end{aligned} \tag{19}$$

For those $i$ similar to $i_1$, we know $u$ likes $i$ because $u$ likes $i_1$. Hence, the only explanation of $\tilde{R}_{ui} = 0$ is that $m_{ui}$ is so small that $u$ misses $i$. The first term in Eq.(19) leads to the same conclusion. To be specific, given $i$ and $i_1$ are similar, $\boldsymbol{\omega_{i_1}^\top \omega_i}$ is positive because $\boldsymbol{\omega_i}$ and $\boldsymbol{\omega_{i_1}}$ are in the same embedding space. Then we know $\frac{\eta \sigma(\bar{R}_{ui}) \sigma(-\bar{R}_{ui})}{(1 - m_{ui} \sigma(\bar{R}_{ui}))^2} \boldsymbol{\omega_{i_1}^\top \omega_i} \sigma(-\bar{R}_{ui_1})$ is positive, therefore this term contributes to the decrease of $\bar{m}_{ui}$. Since all $\bar{m}_{ui}$ share the same distributed representation, the user-item exposure $\bar{m}_{ui}$ where $i$ is unsimiliar to $i_1$ will be updated automatically. The other three terms in Eq. (19) can be analyzed similarly.

## 3 REAL-WORLD EXPERIMENTS

In this section, we conduct experiments on two real-world datasets and compare several state-of-the-art methods with the proposed method Biif. We aim to answer the following two research questions:

- RQ1: How does Biif perform compared with other existing methods?

- RQ2: Is the proposed bi-level optimization framework necessary in Biif?

### 3.1 Experimental Setup

In this subsection, we introduce the datasets, comparison methods, evaluation protocols, and training details.

#### 3.1.1 Datasets

To be best of our knowledge, the Yahoo!R3[1] dataset and the Coat[2] dataset are the only two public datasets that contain users' ratings for randomly selected items, and we use the two datasets to measure the true recommendation performance of Biif and the comparison methods.

To be specific, the Yahoo dataset contains approximately 15.4k users, 1k songs, and 300k five-star user-song ratings in the training set. Besides, the unbiased test set is collected by sampling a subset of 5,400 users and

---

[1]https://webscope.sandbox.yahoo.com/
[2]https://www.cs.cornell.edu/ schnabts/mnar

Table 1: (RQ1): Experiment Results on Yahoo for Comparison.

| Metrics | RelMF | ExpoMF | CJMF | AD | CEB | BPR | UMF$_{(ours)}$ | Biif$_{(ours)}$ |
|---|---|---|---|---|---|---|---|---|
| DCG@1 | $0.501 \pm 0.004$ | $0.521 \pm 0.007$ | $0.535 \pm 0.003$ | $0.531 \pm 0.006$ | $0.462 \pm 0.005$ | $0.534 \pm 0.007$ | $0.541 \pm 0.009$ | $\mathbf{0.552 \pm 0.004}$ |
| DCG@2 | $0.686 \pm 0.002$ | $0.724 \pm 0.011$ | $0.742 \pm 0.002$ | $0.738 \pm 0.005$ | $0.664 \pm 0.008$ | $0.740 \pm 0.007$ | $0.731 \pm 0.010$ | $\mathbf{0.766 \pm 0.004}$ |
| DCG@3 | $0.794 \pm 0.007$ | $0.849 \pm 0.008$ | $0.866 \pm 0.001$ | $0.852 \pm 0.007$ | $0.779 \pm 0.007$ | $0.856 \pm 0.005$ | $0.842 \pm 0.014$ | $\mathbf{0.888 \pm 0.002}$ |
| MAP@1 | $0.501 \pm 0.004$ | $0.521 \pm 0.007$ | $0.532 \pm 0.007$ | $0.531 \pm 0.006$ | $0.460 \pm 0.005$ | $0.534 \pm 0.009$ | $0.541 \pm 0.009$ | $\mathbf{0.552 \pm 0.004}$ |
| MAP@2 | $0.583 \pm 0.004$ | $0.610 \pm 0.009$ | $0.622 \pm 0.001$ | $0.620 \pm 0.004$ | $0.558 \pm 0.006$ | $0.622 \pm 0.007$ | $0.625 \pm 0.008$ | $\mathbf{0.642 \pm 0.003}$ |
| MAP@3 | $0.608 \pm 0.005$ | $0.636 \pm 0.007$ | $0.646 \pm 0.001$ | $0.643 \pm 0.005$ | $0.586 \pm 0.005$ | $0.645 \pm 0.006$ | $0.652 \pm 0.008$ | $\mathbf{0.664 \pm 0.003}$ |

asking every one of them to rate 10 randomly selected songs. The Coat dataset contains approximately 290 users, 300 coats, and 6,500 five-star user-coat ratings in the training set. Similar to the Yahoo dataset, the unbiased test set is collected by asking all users to rate 16 randomly selected coats.

Both datasets use the following preprocessing procedure. Suggested by Yang et al. (2018), we treat ratings $\geq 4$ as positive feedback and others as negative feedback. We first select the most popular negative item and the most popular positive item for the most active 20% users to form the validation set, which can be approximated as unbiased since the items are very likely to be exposed to the users. Following the suggestion in Ren et al. (2018), we add the unbiased validation set into the training set since the model can always leverage more information from the unbiased validation set. Apart from the unbiased validation set, we select 10% data from the training set to form a hyper-validation set to tune hyperparameters.

### 3.1.2 Comparison methods

We mainly compare Biif with the following methods. **RelMF** Saito et al. (2020) adopts an unbiased estimator and uses the item popularity to approximate exposure; **ExpoMF** Liang et al. (2016b) introduces exposure variables to build a probabilistic model and uses the Expectation-Maximization algorithm to estimate exposure; **CJMF**Zhu et al. (2020) leverages different parts of the training dataset to jointly train multiple models for exposure estimation. Note that Biif and CJMF both adopt the cross-entropy loss in this paper for a fair comparison; **AD**Chen et al. (2021) proposes a general learning framework to debias recommendation and improve relevance; **CEB**Gupta et al. (2021) propose a novel loss function for learning the exposure and correcting exposure bias; **BPR** Rendle et al. (2009) is the most widely pairwise algorithm for the top-N recommenders in implicit feedback; **UMF** uses the same exposure estimation as that in RelMF but adopts our unbiased low-variance estimator.

### 3.1.3 Evaluation protocols

Suggested by Saito et al. (2020), we report the DCG@K (Discounted Cumulative Gain) and MAP@K (Mean Average Precision) to evaluate the ranking performance of all methods. We set $K$=1,2,3 in our experiments since the number of exposed items in the test set is small: Yahoo has 10 items and Coat has 16 items.

### 3.1.4 Training details

We use Pytorch to implement Biif and optimize it with AdamKingma & Ba (2014). We set the learning rate as $10^{-3}$, the hidden dim as 50, the batch size as 1024, the training epoch as 100 for all methods on all datasets unless otherwise specified. Since CJMF uses $C$=8 models, we train CJMF for a total of 15 epochs for a fair comparison. For AD requiring an unbiased validation set, we use the validation set constructed by Biif for a fair comparison. For other hyperparameters such as weight decay, we tune them via the performance on the hyper-validation set using the SNIPS Yang et al. (2018) estimator. We run every experiment five times and report the average result and one standard deviation.

### 3.2 RQ1: Biif outperforms other methods.

In this subsection, we aim to answer RQ1: How does Biif perform compared with other existing methods? Table 1 and Table 2 show the performances for all six methods including Biif on the Yahoo dataset and the Coat dataset respectively.

Table 2: (RQ1): Experiment Results on Coat for Comparison.

| Metrics | RelMF | ExpoMF | CJMF | AD | CEB | BPR | UMF$_{(ours)}$ | Biif$_{(ours)}$ |
|---|---|---|---|---|---|---|---|---|
| DCG@1 | $0.555 \pm 0.013$ | $0.523 \pm 0.033$ | $0.504 \pm 0.022$ | $0.526 \pm 0.009$ | $0.539 \pm 0.014$ | $0.568 \pm 0.015$ | $0.556 \pm 0.013$ | $\mathbf{0.573 \pm 0.015}$ |
| DCG@2 | $0.739 \pm 0.003$ | $0.728 \pm 0.020$ | $0.729 \pm 0.023$ | $0.752 \pm 0.009$ | $0.757 \pm 0.014$ | $0.770 \pm 0.015$ | $0.742 \pm 0.015$ | $\mathbf{0.792 \pm 0.014}$ |
| DCG@3 | $0.887 \pm 0.013$ | $0.851 \pm 0.021$ | $0.873 \pm 0.016$ | $0.907 \pm 0.008$ | $0.899 \pm 0.018$ | $0.906 \pm 0.011$ | $0.896 \pm 0.018$ | $\mathbf{0.931 \pm 0.004}$ |
| MAP@1 | $0.555 \pm 0.013$ | $0.523 \pm 0.033$ | $0.504 \pm 0.022$ | $0.526 \pm 0.009$ | $0.539 \pm 0.014$ | $0.568 \pm 0.015$ | $0.556 \pm 0.013$ | $\mathbf{0.573 \pm 0.015}$ |
| MAP@2 | $0.622 \pm 0.008$ | $0.594 \pm 0.024$ | $0.589 \pm 0.017$ | $0.607 \pm 0.011$ | $0.620 \pm 0.011$ | $0.635 \pm 0.010$ | $0.621 \pm 0.015$ | $\mathbf{0.648 \pm 0.015}$ |
| MAP@3 | $0.636 \pm 0.008$ | $0.606 \pm 0.024$ | $0.603 \pm 0.013$ | $0.626 \pm 0.011$ | $0.638 \pm 0.009$ | $0.650 \pm 0.008$ | $0.639 \pm 0.015$ | $\mathbf{0.659 \pm 0.013}$ |

Firstly, we observe Biif achieves the best performance among all methods on the two datasets. This verifies the effectiveness and robustness of the proposed Biif. Furthermore, Biif outperforms UMF in DCG@3 by about 4% in both datasets because Biif connects exposure estimation not only with the item information but also with the user information.

Last but not least, UMF outperforms RelMF in DCG@3 by about 7.4% in the Yahoo dataset. This can be explained by the high gradient variance problem of RelMF. High gradient variance causes inaccurate gradient updates and thus reduces the recommendation performance. Note that we cannot visualize the gradient variance since the gradient variance comes from the assumption of the randomness of the dataset. A single dataset can be seen as a single data point and thus cannot compute its variance. The advantage of UMF over RelMF in the Coat dataset is smaller. The reason may be that the size of the Coat dataset is small and can not reveal the difference between UMF and RelMF. Note that the high gradient variance problem may also explain why Biif outperforms AD.

As for the time complexity, in the Yahoo dataset, RelMF takes 1.35 seconds to finish one epoch on the GeForce GTX1650 platform and Biif takes 6.54 seconds due to the bi-level optimization computing process. Also, we have done some experiments, which apply clipping parametersSaito et al. (2020) such as 0.01 on RelMF, CJMF, UMF, CEB and Biif. Results are very similar to the ones without clipping parameters and the reason may be reducing variance while introducing bias.

### 3.3 RQ2: Necessity of bi-level optimization

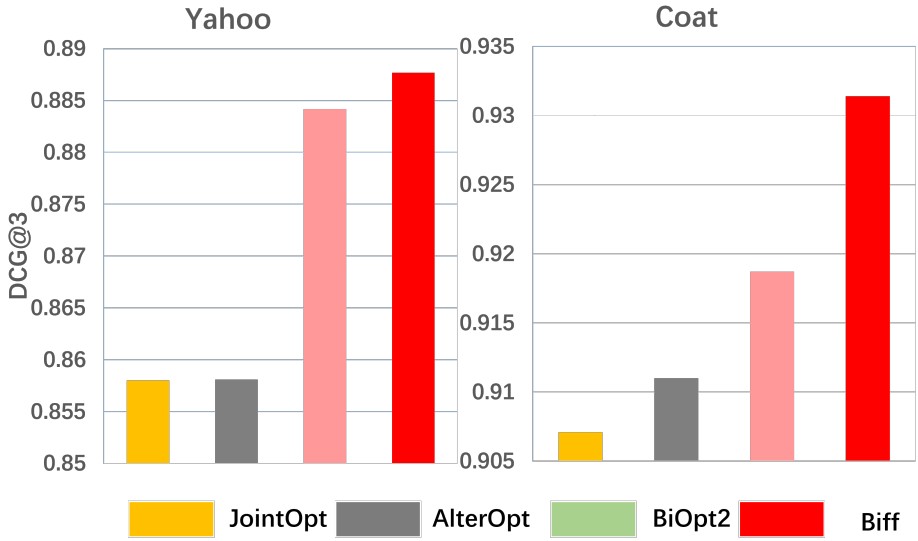

Figure 1: (RQ2): Experiment results on Yahoo and Coat for comparison. This demonstrates the importance of our bi-level framework.

In this subsection, we aim to answer RQ2: Is bi-level optimization necessary in our method? To better understand the necessity of bi-level optimization in Biif, we investigate two baseline strategies, where the

exposure parameters and the relevance parameters are jointly optimized and alternately optimized, respectively. We denote the two baseline strategies as **JointOpt** and **AlterOpt** respectively. As we can see in Fig. 1, JointOpt and AlterOpt yield similar results, and Biif outperforms both of them in DCG3 by around 3% in both datasets. The reason is that JointOpt and AlterOpt do not leverage the information of the unbiased validation set when updating the exposure parameters.

Inspired by Ma et al. (2020), we also consider another baseline strategy of bi-level optimization. Instead of using the unbiased validation set, we treat every train batch as the validation set and perform bi-level optimization between them:

$$\min_{\boldsymbol{\alpha}} \quad \mathcal{L}_{train}(\boldsymbol{\omega}^*(\boldsymbol{\alpha}), \boldsymbol{\alpha}). \tag{1BiOpt2}$$

$$\text{s.t.} \quad \boldsymbol{\omega}^*(\boldsymbol{\alpha}) = \arg\min_{\boldsymbol{\omega}} \mathcal{L}_{train}(\boldsymbol{\omega}, \boldsymbol{\alpha}). \tag{2BiOpt2}$$

Note that exposure in the validation set can not be approximated as 1 anymore, so we use the estimated $\bar{m}_{ui}$ to represent the user-item exposure. We denote this new bi-level optimization strategy as **BiOpt2**. BiOpt2 receives no guidance from the unbiased validation set and thus is worse than Biif. BiOpt2 improves DCG3 over JointOpt and AlterOpt, by around 3% in Yahoo and 1% in coat respectively. The reason may be that BiOpt2 considers the relation between $\boldsymbol{\omega}$ and $\boldsymbol{\alpha}$ explicitly, which narrows down the optimization space to a more reasonable one and thus improves the training process. Similar results are also reported in Ghosh & Lan (2021) where bi-level optimization on a biased validation set achieves satisfying performance.

## 4 SEMI-SYNTHETIC EXPERIMENTS

Besides the real-world dataset experiments which have verified the effectiveness of Biif, we further investigate the correctness of the estimated exposure of Biif on semi-synthetic datasets. Specifically, we aim to investigate the following research question — RQ3: Does Biif learn exposure correctly?

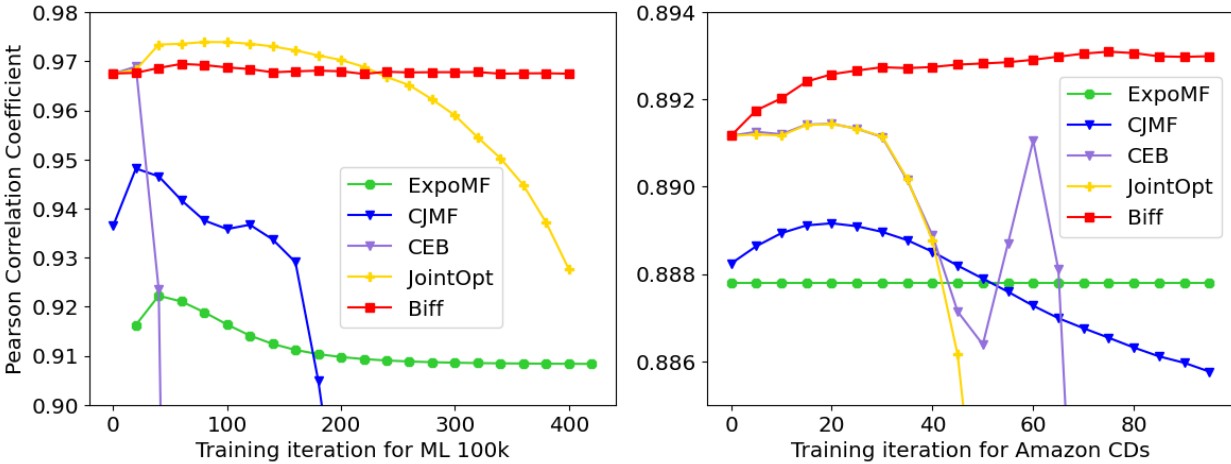

Figure 2: Trend of the PCC value between the ground-truth exposure and the learned exposure for ExpoMF, CJMF, CEB, JointOpt, and Biif on the ML 100K dataset and the Amazon CDs dataset. The Biif line is high and stable all the time, which is consistent with the expressive results of Biif in Fig. 3.

### 4.1 Datasets

To answer RQ3, we need to know the ground-truth exposure parameters in the dataset. We use the MovieLens (ML) 100K[3] dataset and the Amazon CDs[4] dataset to construct semi-synthetic datasets. The ML 100K

---

[3]https://grouplens.org/datasets/movielens/100K/
[4]http://snap.stanford.edu/data/amazon/

dataset is collected by a movie website and contains five-star movie ratings for 1683 movies by 944 users. The Amazon CDs dataset has $3,749,004$ five-star ratings for $486,360$ items by $1,578,597$ users. Following Schnabel et al. (2016); Saito et al. (2020), we create two semi-synthetic datasets based on the Amazon CDs dataset and the ML 100K dataset respectively. For the Amazon CDs dataset, we first remove the users who have less than 10 interactions and the items which have less than 8 interactions. Then we only keep the first 3,000 users and 3,000 items for the Amazon CDs dataset, due to the limited memory of our computer. Specifically, to construct the semi-synthetic dataset, we need to recover the matrix, whose size is the product of the number of the user and the number of the item. Yet the size of the Amazon CDs dataset is too large for the construction process. We do not reduce the size of the ML 100K dataset since the size already fits our computer. The following procedure has three steps:

- $O_{ui}$ is 0 if the rating of $(u, i)$ is observed and 0 otherwise. We use logistic matrix factorization Johnson (2014) to fit $O_{ui}$, and then treat the returned value $\bar{O}_{ui}$ as true exposure parameter $m_{ui}$.

- $R_{ui}$ is 1 if the rating of $(u, i)$ is larger than 3 and 0 otherwise. Similar to $O_{ui}$, we can get true relevance parameter by fitting $R_{ui}$, and treat the returned value $\bar{R}_{ui}$ as the true relevance $\gamma_{ui}$.

- We generate the observe variable $\tilde{R}_{ui}$ as follows: $\tilde{R}_{ui} = R_{ui}O_{ui}$ where $O_{ui} \sim Bernoulli(\theta_{ui})$ and $R_{ui} \sim Bernoulli(\gamma_{ui})$.

The preprocessing procedure of semi-synthetic datasets is the same as that of the Yahoo dataset.

### 4.2 Training and Evaluation

Denote $\bar{m}_{ui}$ as the estimated exposure between $u$ and $i$. To measure the correlation between the estimated exposure $\bar{m}_{ui}$ and the true exposure $m_{ui}$, we introduce Pearson Correlation Coefficient (PCC) Wright (1921). The PCC value ranges from $-1$ to 1. A value approximating to 1 means a strong positive linear relationship between the two variables, and a value approximating to $-1$ means a strong negative linear relationship. A zero value means no linear correlation between the two variables. Note that any metric that can measure the relationship between the estimated exposure and true exposure can also be used. For every user $u$, we compute the PCC value between $\bar{m}_{ui}$ and $m_{ui}$ against all $M$ items:

$$pcc_u = \frac{M \sum_i m_{ui}\bar{m}_{ui} - \sum_i m_{ui} \sum_i \bar{m}_{ui}}{\sqrt{M \sum_i m_{ui}^2 - (\sum_i m_{ui})^2}\sqrt{M \sum_i \bar{m}_{ui}^2 - (\sum_i \bar{m}_{ui})^2}}. \tag{20}$$

We report the average $pcc_u$ for all users. To speed up training in the Amazon CDs, we use a batch size 8096 instead of 1024. The rest of the procedure is similar to that of the real-world experiments.

### 4.3 RQ3: Does Biif learn exposure correctly?

In experiments, we find the performance of JointOpt is very similar to that of AlterOpt so we only report the results of JointOpt. We analyze the PCC value for ExpoMF, CJMF, CEB, JointOpt, and Biif since only the five methods estimate exposure during training. For the Amazon CDs dataset, we find the exposure estimated by ExpoMF barely changed in the whole training process and the exposure updating frequency of ExpoMF is much lower. To better visualize the trend for all four methods, we only plot the PCC line for the first 100 iterations and use a straight line with the mean PCC value to represent the PCC line of ExpoMF. For CJMF, we use the official code in Zhu et al. (2020) and average the estimated exposure on the $C = 8$ models as the final estimated exposure.

**Performance**. In Fig. 3 we observe that Biif is still the best performing method in the Amazon CDs dataset and outperforms other methods except for CJMF on the ML 100K dataset. One possible explanation is that CJMF leverage $C = 8$ models and one residual component simultaneously, which improves training. UMF achieves better performances than RelMF on both datasets due to the low gradient variance.

**PCC Trend Analysis**. Firstly, the PCC values in Fig. 2 for all four methods are larger than 0.8 at early training iterations, which indicates a strong positive linear correlation. This means all methods can estimate

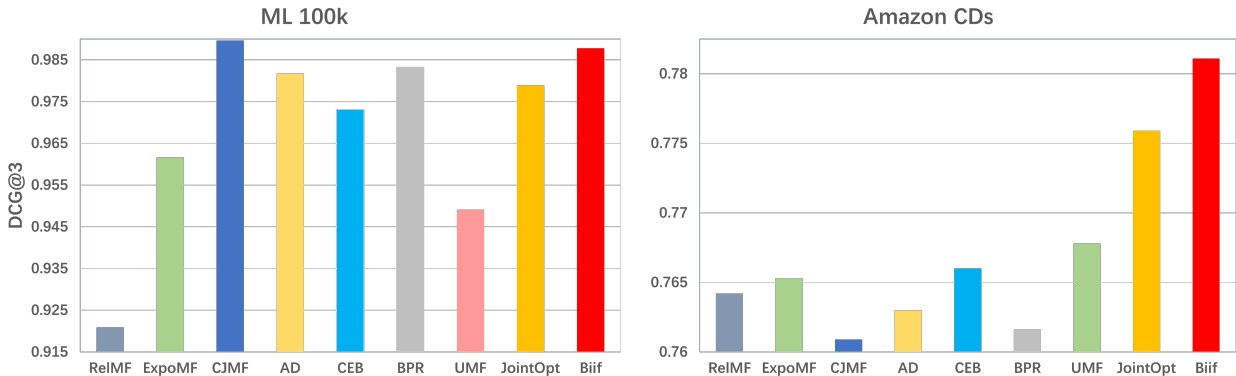

Figure 3: (RQ3): Experiment results (DCG@3) for all comparision methods on both the ML 100K dataset and the Amazon CDs dataset. Biif still performs the best in the Amazon CDs dataset and outperforms other methods except for CJMF on the ML 100K dataset. One possible explanation is that CJMF leverage $C = 8$ models and one residual component simultaneously, which improves training.

exposure correctly. Besides, we observe the ExpoMF line is the lowest at the early training iterations and this is consistent with the unsatisfying results of ExpoMF in Fig. 3. The reason may be that ExpoMF is a biased estimator Saito et al. (2020).

Furthermore, we compare JointOpt with Biif. In Amazon CDs, the Biif line is higher than JointOpt all the time, in accordance with that Biif outperforms JointOpt in Fig. 3. For ML 100K, although the PCC line of JointOpt is higher than Biif at the early training iterations, the JointOpt line experiences a gradual decrease. In real-world datasets, we do not have access to the PCC value which relies on the ground-truth exposure so we can not stop the training process early to get a good result of JointOpt. After some training iterations, the JointOpt line becomes very low in accordance with the results in Fig. 3. The explanation may be that JointOpt updates exposure parameters and relevance parameters on the training set simultaneously, and thus experiences instability during training. In contrast, guided by a small unbiased validation set, Biif can enjoy a stable training process and thus estimate exposure more accurately than JointOpt. The comparison between CEB and Biff can be analyzed similarly. CEB proposes novel loss functions to minimize the true risk objective with the high probability and avoid the trivial solutions, but CEB does not receive the guidance from the unbiased validation set, which explains its unstable exposure learning process.

Last but not least, we make a comparison between CJMF and Biif. In Amazon, the low PCC line of CJMF in Fig. 2 corresponds to the unsatisfying results in Fig. 3. Yet, on ML 100K, CJMF outperforms Biif in terms of performance in Fig. 3 while the PCC line of CJMF is lower than that of Biif in Fig. 2. The reason may be CJMF leverages an extra residual component to improve training, which is not included in the exposure estimation process in Fig. 2.

## 5 RELATED WORK

Our proposed algorithm Biif is inspired by two lines of research in recommender systems: 1) debiasing and 2) bi-level optimization, as illustrated below.

### 5.1 Debiasing

Many important work Steck (2010); Harald (2013); Hernández-Lobato et al. (2014); Wang et al. (2018; 2019b); Joachims & Swaminathan (2016); Wang et al. (2020); Liang et al. (2016a); Bonner & Vasile (2018); Schnabel et al. (2016) have studied the bias in the explicit rating data. For example, as the user can choose which items to rate freely, the observed ratings cannot serve as a representative sample of all ratings. Thus

the biased rating data leads to challenges for both recommendation evaluation and training. To correct this bias, many methods Wang et al. (2018; 2019b); Joachims & Swaminathan (2016); Wang et al. (2020); Liang et al. (2016a) use causal inference to learn from biased data and achieve better performances.

Compared with explicit feedback, implicit feedback is much easier to collect and thus plays a more important role, which renders debiasing in implicit feedback an important topic. The work Yang et al. (2018) develops an unbiased offline evaluator which significantly reduces the bias toward popular items To debias in model training, many methods Hu et al. (2008); Devooght et al. (2015) adopt a heuristic-based strategy, where unobserved interactions are assigned with a lower weight. Furthermore, the methods in Pan et al. (2008); Pan & Scholz (2009) associate the weight with the user's activity and the methods in He et al. (2016); Yu et al. (2017) specify the weight with the item popularity. The work Tran et al. (2021) designs an unbiased learning to rank toolbox on implicit feedback for researchers. CEB Gupta et al. (2021) proposes to leverage known exposure probabilities to mitigate exposure bias for link prediction. From a casual perspective, ExpoMF Liang et al. (2016b) directly incorporates exposure into collaborative filtering and builds a probabilistic model. Based on the IPS technique, RelMF Saito et al. (2020) propose an unbiased estimator with the item popularity as exposure estimation. For better exposure estimation, CJMF Zhu et al. (2020) propose a combinatorial joint learning framework to solve the estimation-training overlap problem. The estimated exposure can still be biased since it only leverages biased training data. Besides, we find the unbiased estimator in Saito et al. (2020); Zhu et al. (2020) suffers from the high gradient variance problem. In this paper, we propose an unbiased estimator with low variance from a probabilistic view. The work Yu et al. (2020) propose to use influence function to correct the data bias, and AD Chen et al. (2021) leverages another set of data to debias data by solving the bi-level optimization problem. The main differences between AD and Biif are a) Biif has a low-variance unbiased estimator while AD does not and b) AD requires an unbiased set in advance at the expense of degrading user experience by deploying random policy on the recommender platform, while Biif forms the unbiased set from the offline biased training set, which is much more practical in real-world scenarios.

### 5.2 Bi-level Optimization

Bi-level optimization Colson et al. (2007) is a mathematical framework where one problem is nested in another. Recently, many important workRendle (2012); Chen et al. (2019); Ma et al. (2020); Lee et al. (2019); Lu et al. (2020) connect recommender system with bi-level optimization due to the nested nature of their formulations. These work treat model parameters as the inner variable and specify the outer variable according to the specific problem. Specifically, the methods Rendle (2012); Chen et al. (2019) use the validation set to update the regularization coefficient during training, where the regularization coefficient serves as the outer variable. Similarly, the work Ma et al. (2020) view the margin in the hinge loss as the learnable parameter and generate different margins for different training triplets. Another research line Lee et al. (2019); Lu et al. (2020) is to treat the model parameters as both the inner variable and the outer variable following Finn et al. (2017). In this work, we view the exposure parameters as the outer variable and the relevance parameters as the inner variable. We first construct an unbiased validation set from the biased training set and then propose to update the exposure parameters with the learning process of the relevance parameters.

## 6 CONCLUSION

To bridge the gap between the implicit feedback and the user-item relevance, existing approaches explicitly model the user-item exposure while the proposed unbiased estimators suffer from high gradient variance. In this paper, we propose a low-variance unbiased estimator from a probabilistic view and this estimator effectively bounds the gradient variance. Besides, we connect exposure estimation with both user and item information and then collect an unbiased set to guide exposure estimation. By leveraging the unbiased set, we update exposure parameters and relevance parameters simultaneously via bi-level optimization. Experiments on real-world datasets and semi-synthetic datasets verify the effectiveness of Biff.

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

## A  Appendix

### A.1  Unbiased Estimator

Our estimator is defined as:

$$\mathcal{L}_2(\boldsymbol{\omega}) = - \sum_{(u,i)\in\mathcal{D}} \tilde{R}_{ui} \log(\bar{m}_{ui}p_{ui})$$
$$+ (1 - \tilde{R}_{ui}) \log(1 - \bar{m}_{ui}p_{ui}). \tag{21}$$

With this definition, we can calculate the expectation of $\mathcal{L}_2(\boldsymbol{\omega})$ as the following:

$$E(\mathcal{L}_2(\boldsymbol{\omega})) = - \sum_{(u,i)\in\mathcal{D}} m_{ui}\gamma_{ui} \log(\bar{m}_{ui}p_{ui})$$
$$+ (1 - m_{ui}\gamma_{ui}) \log(1 - \bar{m}_{ui}p_{ui}). \tag{22}$$

Given an accurate exposure estimation $\bar{m}_{ui} = m_{ui}$, we know the optimal solution which minimizes $E(\mathcal{L}_2(\boldsymbol{\omega}))$ is $p_{ui} = \gamma_{ui}$, which proves our estimator unbiased.

### A.2  Zero Gradient

For analysis convenience, we treat every $\bar{m}_{ui}$ as a learnable parameter instead of using MLP to parameterize $\bar{m}_{ui}$. Recall our estimator is defined as:

$$\mathcal{L}_2(\boldsymbol{\omega}) = - \sum_{(u,i)\in\mathcal{D}} \tilde{R}_{ui} \log(\bar{m}_{ui}p_{ui})$$
$$+ (1 - \tilde{R}_{ui}) \log(1 - \bar{m}_{ui}p_{ui}). \tag{23}$$

For $\tilde{R}_{ui} = 1$, we can write the loss function term related to $u$ and $i$ as:

$$\mathcal{L}_{ui} = - \log(\bar{m}_{ui}p_{ui})$$
$$= - \log(\bar{m}_{ui}) - \log(p_{ui}). \tag{24}$$

Note that the updated $\boldsymbol{\omega_u}$ and $\boldsymbol{\omega_i}$ do not contain any $\bar{m}_{ui}$ term since the $-\log(\bar{m}_{ui})$ term in Eq. (24) is a constant in the inner loop update. As a result, $\mathcal{L}_{val}$ does not contain any $\bar{m}_{ui}$ term, which means:

$$\frac{\partial \mathcal{L}_{val}}{\partial \bar{m}_{ui}} = 0. \tag{25}$$

### A.3  Gradient Analysis

Note that the training loss function with regard to $\bar{m}_{ui}$ for $\tilde{R}_{ui} = 0$ can be written as:

$$\mathcal{L}_{ui} = - \log(1 - \bar{m}_{ui}\sigma(\bar{R}_{ui})). \tag{26}$$

where $\bar{R}_{ui} = \boldsymbol{\omega_u}^\top \boldsymbol{\omega_i}$. After a gradient descent step, the user embedding is updated as:

$$
\begin{aligned}
\boldsymbol{\omega}_u(\bar{m}_{ui}) =& \boldsymbol{\omega_u} - \eta \frac{\partial \mathcal{L}_{ui}}{\partial \boldsymbol{\omega_u}} \\
=& \boldsymbol{\omega_u} - \eta \frac{\partial \mathcal{L}_{ui}}{\partial \sigma(\bar{R}_{ui})} \frac{\partial \sigma(\bar{R}_{ui})}{\partial \boldsymbol{\omega_u}} \\
=& \boldsymbol{\omega_u} - \eta \frac{\bar{m}_{ui} \sigma(\bar{R}_{ui}) \sigma(-\bar{R}_{ui}) \boldsymbol{\omega_i}}{1 - \bar{m}_{ui} \sigma(\bar{R}_{ui})}.
\end{aligned}
\tag{27}
$$

Similarly, the item embedding is updated as:

$$
\boldsymbol{\omega}_i(\bar{m}_{ui}) = \boldsymbol{\omega_i} - \eta \frac{\bar{m}_{ui} \sigma(\bar{R}_{ui}) \sigma(-\bar{R}_{ui}) \boldsymbol{\omega_u}}{1 - \bar{m}_{ui} \sigma(\bar{R}_{ui})}.
\tag{28}
$$

Recall that in the validation set, assume $u$ likes $i_1$ and dislikes $i_2$ ($i_1$ or $i_2$ can not be $i$); $u_1$ likes $i$ and $u_2$ dislikes $i$ ($u_1$ or $u_2$ can not be $u$). We write the loss on the unbiased validation set regard to the related users($u_1$ and $u_2$) and the related items($i_1$ and $i_2$) as:

$$
\begin{aligned}
\mathcal{L}'_{val} = & -\log(\sigma(\bar{R}_{ui_1})) - \log(1 - \sigma(\bar{R}_{ui_2})) \\
& - \log(\sigma(\bar{R}_{u_1 i})) - \log(1 - \sigma(\bar{R}_{u_2 i})).
\end{aligned}
\tag{29}
$$

According to the chain rule, we can write the gradient as:

$$
\frac{\partial L'_{val}}{\partial \bar{m}_{ui}} = \frac{\partial \mathcal{L}'_{val}}{\partial \boldsymbol{\omega_u}^\top} \frac{\partial \boldsymbol{\omega_u}}{\partial \bar{m}_{ui}} + \frac{\partial \mathcal{L}'_{val}}{\partial \boldsymbol{\omega_i}^\top} \frac{\partial \boldsymbol{\omega_i}}{\partial \bar{m}_{ui}}.
\tag{30}
$$

We unroll the $\frac{\partial L'_{val}}{\partial \boldsymbol{\omega_u}^\top}$ term as:

$$
\begin{aligned}
\frac{\partial \mathcal{L}'_{val}}{\partial \boldsymbol{\omega_u}^\top} = & -\frac{\sigma(\bar{R}_{ui_1}) \sigma(-\bar{R}_{ui_1})}{\sigma(\bar{R}_{ui_1})} \boldsymbol{\omega}_{i_1}^\top \\
& -\frac{-\sigma(\bar{R}_{ui_2}) \sigma(-\bar{R}_{ui_2})}{1 - \sigma(\bar{R}_{ui_2})} \boldsymbol{\omega}_{i_2}^\top \\
= & -\sigma(-\bar{R}_{ui_1}) \boldsymbol{\omega}_{i_1}^\top + \sigma(\bar{R}_{ui_2}) \boldsymbol{\omega}_{i_2}^\top.
\end{aligned}
\tag{31}
$$

Similarly, the $\frac{\partial \mathcal{L}'_{val}}{\partial \boldsymbol{\omega_i}}$ term is unrolled as:

$$
\begin{aligned}
\frac{\partial \mathcal{L}'_{val}}{\partial \boldsymbol{\omega_i}^\top} = & -\frac{\sigma(\bar{R}_{u_1 i}) \sigma(-\bar{R}_{u_1 i})}{\sigma(\bar{R}_{u_1 i})} \boldsymbol{\omega}_{u_1}^\top \\
& -\frac{-\sigma(\bar{R}_{u_2 i}) \sigma(-\bar{R}_{u_2 i})}{1 - \sigma(\bar{R}_{u_2 i})} \boldsymbol{\omega}_{u_2}^\top \\
= & -\sigma(-\bar{R}_{u_1 i}) \boldsymbol{\omega}_{u_1}^\top + \sigma(\bar{R}_{u_2 i}) \boldsymbol{\omega}_{u_2}^\top.
\end{aligned}
\tag{32}
$$

We unroll the $\frac{\partial \boldsymbol{\omega_u}(\bar{m}_{ui})}{\partial \bar{m}_{ui}}$ term as:

$$
\begin{aligned}
\frac{\partial \boldsymbol{\omega_u}(\bar{m}_{ui})}{\partial \bar{m}_{ui}} = & -\eta \sigma(\bar{R}_{ui}) \sigma(-\bar{R}_{ui}) \boldsymbol{\omega_i} \frac{\partial \frac{\bar{m}_{ui}}{1 - \bar{m}_{ui} \sigma(\bar{R}_{ui})}}{\partial \bar{m}_{ui}} \\
= & -\frac{\eta \sigma(\bar{R}_{ui}) \sigma(-\bar{R}_{ui}) \boldsymbol{\omega_i}}{(1 - \bar{m}_{ui} \sigma(\bar{R}_{ui}))^2}.
\end{aligned}
\tag{33}
$$

Similarly, the $\frac{\partial \boldsymbol{\omega_i}(\bar{m}_{ui})}{\partial \bar{m}_{ui}}$ term is unrolled as:

$$
\begin{aligned}
\frac{\partial \boldsymbol{\omega_i}(\bar{m}_{ui})}{\partial \bar{m}_{ui}} = & -\eta \sigma(\bar{R}_{ui}) \sigma(-\bar{R}_{ui}) \boldsymbol{\omega_u} \frac{\partial \frac{\bar{m}_{ui}}{1 - \bar{m}_{ui} \sigma(\bar{R}_{ui})}}{\partial \bar{m}_{ui}} \\
= & -\frac{\eta \sigma(\bar{R}_{ui}) \sigma(-\bar{R}_{ui}) \boldsymbol{\omega_u}}{(1 - \bar{m}_{ui} \sigma(\bar{R}_{ui}))^2}.
\end{aligned}
\tag{34}
$$

In the end, we write $\frac{\partial \mathcal{L}'_{val}}{\partial \bar{m}_{ui}}$ as:

$$
\begin{aligned}
\frac{\partial \mathcal{L}'_{val}}{\partial \bar{m}_{ui}} &= \frac{\partial \mathcal{L}'_{val}}{\partial \boldsymbol{\omega}_{\boldsymbol{u}}^\top} \frac{\partial \boldsymbol{\omega}_{\boldsymbol{u}}(\bar{m}_{ui})}{\partial \bar{m}_{ui}} + \frac{\partial \mathcal{L}'_{val}}{\partial \boldsymbol{\omega}_{\boldsymbol{i}}^\top} \frac{\partial \boldsymbol{\omega}_{\boldsymbol{i}}(\bar{m}_{ui})}{\partial \bar{m}_{ui}} \\
&= [\sigma(-\bar{R}_{ui_1})\boldsymbol{\omega}_{i_1}^\top - \sigma(-\bar{R}_{ui_2})\boldsymbol{\omega}_{i_2}^\top] \frac{\eta\sigma(\bar{R}_{ui})\sigma(-\bar{R}_{ui})}{(1 - \bar{m}_{ui}\sigma(\bar{R}_{ui}))^2} \boldsymbol{\omega}_{\boldsymbol{i}} \\
&\quad + [\sigma(-\bar{R}_{u_1 i})\boldsymbol{\omega}_{u_1}^\top - \sigma(-\bar{R}_{u_2 i})\boldsymbol{\omega}_{u_2}^\top] \frac{\eta\sigma(\bar{R}_{ui})\sigma(-\bar{R}_{ui})}{(1 - \bar{m}_{ui}\sigma(\bar{R}_{ui}))^2} \boldsymbol{\omega}_{\boldsymbol{u}} \\
&= \frac{\eta\sigma(\bar{R}_{ui})\sigma(-\bar{R}_{ui})}{(1 - \bar{m}_{ui}\sigma(\bar{R}_{ui}))^2} [\boldsymbol{\omega}_{i_1}^\top \boldsymbol{\omega}_{\boldsymbol{i}}\sigma(-\bar{R}_{ui_1}) \\
&\quad -\boldsymbol{\omega}_{i_2}^\top \boldsymbol{\omega}_{\boldsymbol{i}}\sigma(\bar{R}_{ui_2}) + \boldsymbol{\omega}_{u_1}^\top \boldsymbol{\omega}_{\boldsymbol{u}}\sigma(-\bar{R}_{u_1 i}) \\
&\quad -\boldsymbol{\omega}_{u_2}^\top \boldsymbol{\omega}_{\boldsymbol{u}}\sigma(\bar{R}_{u_2 i})].
\end{aligned} \tag{35}
$$

By analyzing this equation, we can have a natural interpretation of the bi-level optimization framework in Biif as we have discussed in Sec 2.3.2. This also demonstrates the effectiveness of the proposed bi-level framework.

