# OpenReview forum: "A Bi-level Framework for Debiasing Implicit Feedback with Low Variance"
_TMLR — Rejected by TMLR_

### Review · Reviewer_X4br · 2023-02-06

**Summary Of Contributions:**

This paper tackles the biased implicit bias problem in recommender system datasets. The paper argues that existing methods have the high gradient variance problem, then propose a new formulation that has a lower variance, which involves the estimation of exposure. The paper then parameterizes the exposure, and use bi-level optimization. Experiments are conducted offline on standard settings.

**Audience:**

Yes

**Claims And Evidence:**

No

**Requested Changes:**

I believe the authors should use \citep in most places.

Why use m for O?

Close comparison with the AD method.

Overall the paper needs more convincing arguments to claim novelty, and concreteness of the experiments.


**Strengths And Weaknesses:**

Strengths

The paper studies an important problem.

The basic formulation looks simple and looks right to the reviewer.


Weakness

Please be careful about the editing. The paper is messy now and is not readable in many places.

One contribution of this work is that the validation set is not gathered from a random policy. The paper proposes to use "the most popular positive item and negative item for each active user". The reviewer is not convinced that it is sensible - heuristic is heuristic. This is especially for unbiased learning where unbiaseness is critical.

The major contribution of this work is to reduce variance. It only addresses the variance problem with respect to m, which looks incremental.

Related work looks outdated. Difference from existing work look incremental.

- AutoDebias: Learning to Debias for Recommendation

Difference is incremental - this paper also uses bi-level optimization. In fact, bi-level optimization is quite common that it is not clear if it is really a contribution and should be highlighted in the title.

It is not clear how the proposed method can outperform it. Being a very relevant approach, it needs closer examination, instead of hidden in the tables. For example, AD needs a holdout validation dataset, this may not be ideal practically, but performance-wise it should help.

- Neutralizing Popularity Bias in Recommendation Models

This is just an example of more recent work. There are many more out there. Most references date back to 2020, which is not good for this evolving field.

- KDCRec: Knowledge Distillation for Counterfactual Recommendation Via Uniform Data
-- A relevance work that is not discussed.

- Bilateral self-unbiased learning from biased implicit feedback. SIGIR2022
-- Also address the variance problem. No discussion against it.

---

### Review · Reviewer_8XS4 · 2023-02-19

**Summary Of Contributions:**

In this paper, the authors study the problem of unbiased learning in the recommendation. To address this problem, the authors propose a bi-level framework for de-biasing implicit feedback with low variance. Specifically, the authors parameterize the user-item exposure by incorporating both user and item information and propose to construct the unbiased validation set only from the biased training set instead of using a random policy. The experimental results on several datasets demonstrate the efficacy of the proposed approach.

**Audience:**

Yes

**Claims And Evidence:**

Yes

**Requested Changes:**

Please see the Strengths And Weaknesses

**Strengths And Weaknesses:**

Strengths:
1. This paper is well-written and easy to follow.
2. The studied problem is highly significant.
3. The experiments conducted in this paper appear rigorous.

Weaknesses:
1. The problem setting is not novel, as previous works have explored debiasing recommender systems with selection bias extensively. Therefore, it would be useful to clarify how this work builds on previous research in this area.
2. While the authors' technical contributions are sound, they are somewhat limited since the proposed bi-level optimization strategy has been proposed in previous works [1,2,3]. It would be helpful to describe how the authors' approach differs from previous proposals, and how it improves upon them.
3. The proposed unbiased estimator lacks a detailed theoretical analysis. The expectation of the unbiased estimator should be over the exposure variable and the proposed estimator is not theoretically unbiased.
4. The justification for selecting the most popular positive and negative items for each active user to form the unbiased validation set is not clear. The reviewer believes that this is a key component of the paper, but its theoretical underpinnings require further elaboration.
5. Some important baseline methods, such as KD-Label [4] and CausE [5], are missing from the experiments, which could limit the strength of the conclusions drawn from this work. Including these baseline methods in the experiments would provide a more comprehensive evaluation of the proposed approach.

[1] Learning to Reweight Examples for Robust Deep Learning. ICML 2018
[2] Meta-Weight-Net: Learning an Explicit Mapping For Sample Weighting. NeurIPS 2019.
[3] AutoDebias: Learning to Debias for Recommendation. SIGIR 2021.
[4] A General Knowledge Distillation Framework for Counterfactual Recommendation via Uniform Data. SIGIR 2020
[5] Causal embeddings for recommendation. RecSys 2018.

---

> ### Author Response · Authors · 2023-03-02
> **Response to review questions**
>
> ### General Reply
>
> Many thanks for your valuable and constructive comments on clarifying, correcting, and improving the materials in this paper! We will carefully revise the paper according to your comments as explained below.
>
>
> ### Weaknesses
>
> > The problem setting is not novel, as previous works have explored debiasing recommender systems with selection bias extensively. Therefore, it would be useful to clarify how this work builds on previous research in this area.
>
> We have discussed the related work of debiasing in Sec 5 RELATED WORK.  In this paper, we propose an unbiased estimator with low variance from a probabilistic view. The work Yu et al. (2020) propose to use influence function to correct the data bias, and AD Chen et al. (2021) leverages another set of data to debias data by solving the bi-level optimization problem. The main differences between influence function/AD and Biif are a) Biif has a low-variance unbiased estimator while influence function/AD does not and b) influence function/AD requires an unbiased set in advance at the expense of degrading user experience by deploying random policy on the recommender platform, while Biif forms the unbiased set from the offline biased training set, which is much more practical in real-world scenarios.
>
> > While the authors' technical contributions are sound, they are somewhat limited since the proposed bi-level optimization strategy has been proposed in previous works [1,2,3]. It would be helpful to describe how the authors' approach differs from previous proposals, and how it improves upon them.
>
> Like we discussed the related work section, previous bi-level optimization frameworks like influence function/AD require an unbiased set in advance at the expense of degrading user experience by deploying random policy on the recommender platform, while our proposed Biif forms the unbiased set from the offline biased training set, which is much more practical in real-world scenarios.
>
> > The proposed unbiased estimator lacks a detailed theoretical analysis. The expectation of the unbiased estimator should be over the exposure variable and the proposed estimator is not theoretically unbiased.
>
> We have provided a detailed analysis of the unbiased estimator in Appendix A.1. We illustrate the unbiasedness of the estimator theoretically by showing we will obtain the unbiased relevance.
>
> > The justification for selecting the most popular positive and negative items for each active user to form the unbiased validation set is not clear. The reviewer believes that this is a key component of the paper, but its theoretical underpinnings require further elaboration.
>
> This unbiased set is different from the uniform set colleted by deploying random policy. In the uniform set, we know the item is exposed to the user and thus the interaction is the same as the relevance. In our unbiased set, with the most popular positive and negative items for each active user, the exposure of the user-item pair should be large, and thus we approximate the user-item exposure as 1. This can provide weak supervison signal to update the exposure parameters of the training set.
>
> > Some important baseline methods, such as KD-Label [4] and CausE [5], are missing from the experiments, which could limit the strength of the conclusions drawn from this work. Including these baseline methods in the experiments would provide a more comprehensive evaluation of the proposed approach.
>
> Thanks for pointing out the baseline methods KD-Label [4] and CausE [5] ! While we are aware of these methods, we do not compare our method with them since both KD-Label and CausE require a uniform set, while there is no uniform set in our problem setting.

---

### Review · Reviewer_5cAE · 2023-03-13

**Summary Of Contributions:**

In this work authors investigated a low-variance estimator for the implicit feedback setting. To solve the high-variance suffered by many implicit feedback estimators, authors first proposed a low-variance estimator and then wrap it by a bi-level optimization framework to ensure having an unbiased estimator.

**Audience:**

Yes

**Broader Impact Concerns:**

I do not have concerns on ethical implications of the developed work.

**Claims And Evidence:**

Yes

**Requested Changes:**

I liked the paper, I do not have change requests.

**Strengths And Weaknesses:**

I found the theoretical contribution quite intuitive and solid, a similar reasoning can be found in many online learning works which might give authors ideas for extension. That being said I think it addresses an interesting research question which is also practically relevant. I found the paper well written and the results are described clearly. The proposed solution does not have clear disadvantages or issues.

Besides the theoretical aspects they also compare the proposed solution with respect to many existing methods on different real dataset. I found the evaluation approach reliable and well carried.

---

> ### Author Response · Authors · 2023-03-14
> **Response to review questions**
>
> Dear Reviewer,
>
> Thank you for taking the time to review our paper and for providing your valuable feedback. We are pleased to hear that you found our theoretical contribution intuitive and solid, and that the proposed solution was well-received and compared favorably to existing methods.
>
> We appreciate your comment on the potential for extensions of our work, and we will certainly keep this in mind as we continue to explore this area of research. We are also glad to hear that you found our evaluation approach reliable and well-carried.
>
> We would like to express our gratitude for your positive feedback. Thank you again for your time and effort in reviewing our paper.

---

### Review · Reviewer_rmTc · 2023-03-16

**Summary Of Contributions:**

The authors consider the problem of implicit matrix factorization. The authors consider the algorithm presented in "Unbiased recommender
learning from missing-not-at-random implicit feedback" and propose an improvement that changes how the gradient of the loss function is computed so as to reduce its variance when items are not relevant to the user. They demonstrate their method improves upon prior work in both static and semi-synthetic experiments.

**Audience:**

Yes

**Broader Impact Concerns:**

Given the authors propose a method for recommender systems, maybe they could include a broader impact statement since these systems often interact with a large user population.

**Claims And Evidence:**

No

**Requested Changes:**

I think this paper would be much stronger if the authors also compared against the work presented in Collaborative Filtering for Implicit Feedback Datasets by Hu et al. Since RelMF, which the authors improve upon, itself represented a significant improvement over the work of Hu et al., this would contextualize the progress made by the authors' proposed change.

**Strengths And Weaknesses:**

This is a paper that addresses the important and relevant problem of implicit recommendation. I felt like the paper's exposition was also quite clear and the authors did a good job of explaining how their method works.

However, the paper in its current format feels incomplete. In particular, the motivation for the authors' proposed gradient estimator felt lacking. The correction proposed by the authors is specialized to a single algorithm (RelMF) and only solves the variance issue in the case of m being close to 0 but not p. The authors show empirical performance improvements across static and semi-synthetic benchmarks, however, I was left unsure about how significant those improvements were.

---

> ### Author Response · Authors · 2023-03-19
> **Response to review questions**
>
> ## General Reply
>
> Thank you for taking the time to provide us with your valuable and constructive comments. We appreciate your feedback and insights, which will undoubtedly help us to refine, enhance, and strengthen the content of this paper. We will diligently revise the paper in accordance with your comments, as detailed below.
>
>
>
>
> ## Strengths And Weaknesses:
>
>
> > However, the paper in its current format feels incomplete. In particular, the motivation for the authors' proposed gradient estimator felt lacking. The correction proposed by the authors is specialized to a single algorithm (RelMF) and only solves the variance issue in the case of m being close to 0 but not p.
>
> Thank you for raising this important concern. We appreciate the opportunity to clarify our work further. The correction we propose is indeed not limited to RelMF, but rather, it can be applied to any algorithm that is compatible with Eq.(4). In this context, the $p_{ui}$ is not necessarily computed using matrix factorization methods. Our aim is to address the variance issue related to the m component, which we believe is already a substantial improvement in the performance of these algorithms.
>
>
>
>
> > The authors show empirical performance improvements across static and semi-synthetic benchmarks, however, I was left unsure about how significant those improvements were.
>
> Thank you for pointing out the need for clarification regarding the significance of the performance improvements. In our experiments, we performed each test five times and reported both the average result and one standard deviation. As illustrated in the paper, our proposed method demonstrates noticeable improvements compared to the baseline methods, showcasing the effectiveness of our approach.
>
> ## Requested Changes:
> > I think this paper would be much stronger if the authors also compared against the work presented in Collaborative Filtering for Implicit Feedback Datasets by Hu et al. Since RelMF, which the authors improve upon, itself represented a significant improvement over the work of Hu et al., this would contextualize the progress made by the authors' proposed change.
>
> Thank you for your insightful suggestion. We have conducted additional experiments comparing our method with the Weighted Matrix Factorization (WMF) approach presented in "Collaborative Filtering for Implicit Feedback Datasets" by Hu et al. on both the Yahoo and Coat datasets. The results are as follows:
>
> |  Metric  | DCG1  | DCG2  | DCG1 | MAP1 | MAP2 | MAP3
> |:----:|:----:|:----:|:----:|:----:|:----:|:----:|
> | Yahoo | 0.467 $\pm$ 0.005 | 0.670 $\pm$ 0.007|0.780 $\pm$  0.005 |0.467  $\pm$  0.005 |0.572 $\pm$  0.003 |0.591 $\pm$  0.002 |
> | Coat | 0.540 $\pm$  0.012|0.741 $\pm$  0.003 |0.890 $\pm$  0.015 | 0.540 $\pm$  0.012|0.620 $\pm$  0.010 | 0.636 $\pm$ 0.005|
>
>
> WMF's performance is relatively inferior to that of RelMF, while RelMF's performance is somewhat lower than that of our proposed method, Biff. These findings further demonstrate the effectiveness of Biff in comparison to existing methods, and we appreciate the opportunity to enhance our paper by including these additional comparisons.

---

> > ### Author Response · Authors · 2023-03-19
> > **Response to review questions**
> >
> > ## Broader Impact Concerns:
> > > Given the authors propose a method for recommender systems, maybe they could include a broader impact statement since these systems often interact with a large user population.
> >
> > Thank you for your suggestion to include a broader impact statement in our paper. We acknowledge the importance of discussing the broader implications of our proposed method for recommender systems, as they indeed have a significant influence on large user populations. In our revised manuscript, we will include a dedicated section to address the broader impact of our work.
> >
> > In this section, we will discuss the following aspects:
> >
> > Enhanced user experience: Our proposed gradient estimator aims to improve the accuracy and convergence of implicit recommendation algorithms. As a result, users can benefit from more relevant and personalized recommendations, leading to an overall better experience when interacting with the recommender system.
> >
> > Fairness and diversity: We will discuss the potential impact of our method on fairness and diversity in recommendations. While our primary goal is to reduce the variance in gradient estimations, we acknowledge the need to ensure that our approach does not inadvertently perpetuate biases or reduce the diversity of recommendations.
> >
> > Scalability: Our method can be applied to large-scale recommender systems, which are typically characterized by a vast number of users and items. We will address the potential computational and storage implications of our method on such systems and discuss possible strategies for optimization.
> >
> > Ethical considerations: We will discuss potential ethical concerns related to the implementation of our method in recommender systems, such as user privacy, data collection, and the potential for manipulation of recommendations.
> >
> > By incorporating a broader impact statement in our revised manuscript, we aim to provide a comprehensive perspective on the potential consequences of our proposed method for recommender systems and their user populations. We appreciate your valuable feedback and the opportunity to improve our paper.

---

### Decision · Action_Editors · 2023-05-11

**Recommendation:** Reject

**Comment:**

As pointed in the answer to "Claims", the central difficulty is that the methods rely on an unusual way to achieve unbiasedness which is not guaranteed to be unbiased. Yet the work is interesting and I think a few modifications would make it suitable for publication, so I encourage authors to resubmit a version with the following changes:
1. Running experiments with competitors resorting to a uniformly collected dataset without using it
2. Lowering the claims about unbiasedness which is not guaranteed. It would be even better to quantify the maximum bias but it might be tricky.

**Audience:**

All reviewers agreed on the interest of the problem for TMLR, so the main question here is about claims and pieces of evidence.

**Claims And Evidence:**

Reviewer X4br and 8XS4 point out  flaws in the current version of the paper:

1. The proposed unbiased estimator lacks a detailed theoretical analysis raising questions about the validity of the claim for unbiasedness. More precisely the paper claims to lay into the field of unbiased learning to rank, yet unbiasedness is not proven into the paper. Digging into it it turns out that everything holds on the construction of an "unbiased validation set" by "select[ing] the most popular positive item and negative item for each active user" which is in fact, not guaranteed to be unbiased. This is misleading.  The claim for unbiasedness should be lowered everywhere in the paper before acceptance. Yet the idea is interesting and discussing more why this set is better than a uniformly sampled one would be interesting (the paper already mentions that the experience might be better for the user rather than a truly uniform dataset)

2. Some important baseline methods are missing from the experiments which limits the conclusions drawn from this work. In particular, having no citations of papers from the last three years is surprising for an evolving field. This absence of comparison to recent works seems to be tied to the fact that the proposed work does not require a uniform set collected by deploying a random policy. Yet the paper does not focus on this point which is central in my opinion. As they do not require this uniform set, authors could run tests on data where this set is present and report the difference in performance (without tying acceptance to the fact that the method works better since not having access to such uniform data is a drawback).

3. Concerns about the origin of the performance.  Some code is provided, I looked at it and it does not seem to do anything weird so not knowing why it works better is not a blocker even if a discussion would be welcome.